# Composable Part-Based Manipulation

**Weiyu Liu**[1], **Jiayuan Mao**[2], **Joy Hsu**[1], **Tucker Hermans**[3,4], **Animesh Garg**[3,5], **Jiajun Wu**[1]

[1]Stanford  [2]MIT  [3]NVIDIA  [4]University of Utah  [5]Georgia Tech

**Abstract:** In this paper, we propose *composable part-based manipulation* (CPM), a novel approach that leverages object-part decomposition and part-part correspondences to improve learning and generalization of robotic manipulation skills. By considering the functional correspondences between object parts, we conceptualize functional actions, such as pouring and constrained placing, as combinations of different correspondence constraints. CPM comprises a collection of composable diffusion models, where each model captures a different inter-object correspondence. These diffusion models can generate parameters for manipulation skills based on the specific object parts. Leveraging part-based correspondences coupled with the task decomposition into distinct constraints enables strong generalization to novel objects and object categories. We validate our approach in both simulated and real-world scenarios, demonstrating its effectiveness in achieving robust and generalized manipulation capabilities. For videos and additional results, see our website: https://cpmcorl2023.github.io/.

**Keywords:** Manipulation, Part Decomposition, Diffusion Model

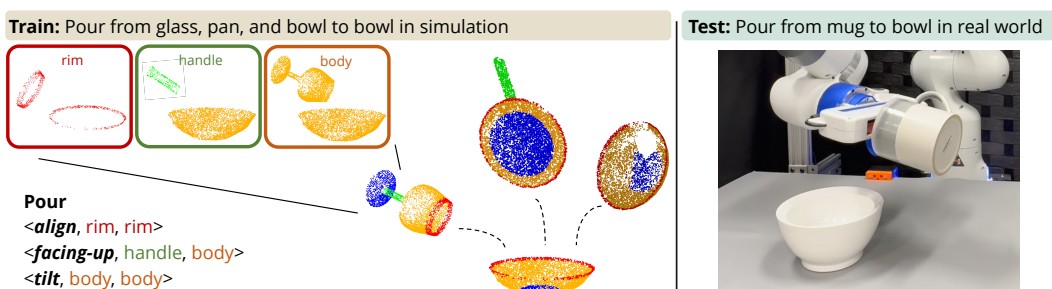

**Figure 1:** CPM composes part-based diffusion models to predict target object poses directly from point clouds. In this example, we show that the "pouring" action is decomposed into three part-based correspondences, which generalize manipulation across object categories, and from simulation to the real world

## 1 Introduction

Compositionality provides appealing benefits in robotic manipulation, as it enables efficient learning, reasoning, and planning. Prior works have extensively studied the decomposition of scenes into objects and their relationships [1, 2, 3], as well as the division of long-horizon plans into primitive skills [3, 4], in order to navigate complex environments and devise long-horizon plans. In this paper, we present a different view of compositionality by considering *object-part decomposition* based on functionality (e.g., *rim, handle, body*), and leverage such decomposition to improve the learning of geometric and physical relationships for robot manipulation.

In the context of language descriptions of objects, part names not only describe the geometric shapes of the parts but also capture their functional affordances. For instance, as depicted in Figure 1, for the action of "pouring", the *rims* define the boundary for alignment between the objects, the *body* of the pouring vessel should be tilted for the action, and its *handle* provides a constraint on the direction the object should face when pouring. Leveraging this knowledge of part affordances, we posit that a family of functional actions, such as pouring and constrained placing, can be conceptualized as a combination of functional correspondences between object parts. Modeling actions using such a

7th Conference on Robot Learning (CoRL 2023), Atlanta, USA.

decomposition yields two important generalizations. First, it enables action generalization to novel instances from the same object category. Second and more importantly, it facilitates generalization to *unseen* object categories. For example, after learning part affordances for the "pouring" action, our robot trained on "pour from *bowls*" and "...*pans*" can generalize to "pour from *mugs*", with no additional training necessary for manipulation with the new object category.

Motivated by these insights, we present the *composable part-based manipulation* (CPM). CPM comprises a collection of diffusion models, where each model captures the correspondence between parts of different objects. These conditional diffusion models take the geometry of the object parts as input and generate parameters for manipulation skills, such as the starting and ending poses of a bowl during the pouring action. Specifically, each model outputs a distribution of feasible trajectories that satisfy a particular correspondence. After learning a collection of composable diffusion models, we represent actions as combinations of part-part correspondences. During inference, we leverage the composition of primitive diffusion models to sample trajectories that adhere to all the part correspondences. This approach improves generalization to novel object categories over models that do not reason about both parts and composable correspondence constraints.

In summary, this paper makes two key contributions. First, we propose composable part-based manipulation, which models manipulation actions as a composition of part-part correspondences between objects. Second, we develop diffusion models trained to capture primitive functional correspondences that can be flexibly recombined during inference. CPM achieves strong generalization across various dimensions, including novel object instances and object categories. We validate the efficacy of CPM on both PyBullet-based simulations and real-robot experiments.

## 2    Related Work

**Object representations for manipulation.**   Prior works use segmentations of common object parts (e.g., blades, lids, and handles) for manipulating articulated objects [5, 6, 7, 8] as well as for transfer to novel objects [9, 10]. A common approach that has been shown effective across different manipulation domains [11, 12, 13] first predicts which part of an object the robot should focus on (e.g., the handle), and then predicts an action relative to the part. Closely related is visual affordance detection [14, 15, 16], which segments objects into different functional regions, such as graspable parts and support surfaces of objects. These functional regions can be shared by more distinct objects, and can be useful for generalizing task-oriented grasping between object categories [17, 18]. Keypoints are another representation that shows robustness to large intra-category shape variation and topology changes [19]. Each keypoint set can provide essential pose information, that lacks in previous segmentation approaches, to support tasks such as hanging mugs on pegs by their handles. The initial supervised approach [19] has been extended to methods that discover keypoints from interactions [20, 21] and from unlabeled videos [22]. Recently, implicit object representations have been used to provide correspondence between any point within the same object category generalizing across 6-DoF pose changes [23, 24, 25]. Large pretrained vision models also support the development of object representations; recent works leverage these models to significantly reduce domain-specific training data, showing strong results for open-vocabulary part segmentation [26], few-shot affordance segmentation [27], and one-shot pose estimation on any novel object from the same category [28]. Despite this huge progress, we still lack object representations that support strong generalization of manipulation to new object categories. We focus on tackling this problem.

**Learning interactions of objects.**   Works in robotics have established the importance of modeling interactions of objects. Recent approaches directly work on 3D observations, without relying on known object models. Learning spatial relations between objects enables the picking and placing of objects at specific locations [1, 29, 30, 2, 31], such as placing an object in the middle drawer, stacking objects, and setting the table. These relations can be extended to represent the logical state of the world to support planning for long-horizon tasks [3, 32, 33]. Other works focus on learning lower-level interactions between objects, such as placing an object stably on a messy tabletop and pushing an object using a tool [34, 35]. For example, O2O-afford [34] correlates feature maps extracted from two objects using a point convolution and outputs a point-wise interaction heatmap.

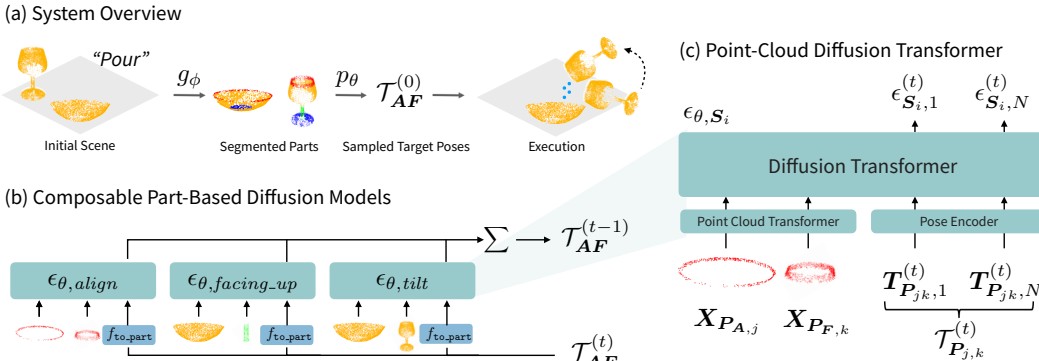

**Figure 2:** (a) Given a task, the partial point clouds of the anchor and function objects, and their parts extracted from a learned segmentation model $g_\phi$, we sample a sequence of transformations from a learned distribution $p_\theta$ to parameterize the function object's trajectory. (b) CPM can be generalized to novel object categories because it decomposes each action to a collection of functional correspondences between object parts. To sample the target transformations that satisfy all functional correspondences, CPM combines the noise predictions from a collection of primitive diffusion models at inference time. (c) Each primitive diffusion model learns a target pose distribution that satisfies a particular part-part correspondence, based on the point clouds of the object parts.

Functionals defined on top of object-wise signed distance functions can also represent constraints on interactions between objects such as contact and containment [36]. Flow-based methods can also learn static relations between objects [37] as well as tool use [38], directly from point clouds. A main difference between our work and these methods is that we bridge the modeling of interactions and object representations through object-part decomposition and learned part-part correspondences, and enjoy empirically validated improvement in generalization.

**Composable diffusion models.** A set of recent works have investigated the potential of diffusion models in robotics [39, 40, 41, 42, 43, 44, 45, 46, 2, 47]. Research demonstrates that diffusion models can generate multimodal distributions over actions [41] and can handle spatial ambiguities in symmetric objects [2]. In image domains, prior work has shown a connection between conditional diffusion models and energy-based models, and proposed techniques to generate images by combining diffusion noises for different language conditions [48]. Recent work provides a more principled way to sample from individually trained models using MCMC [49]. Another approach combines diffusion models by using additional trained adapters for generating faces [50]. CPM combines both lines of work to propose composable diffusion models for robotic manipulation. In doing so we must address two challenges of adapting diffusion models to (1) output poses instead of pixels and (2) combine actions in different part frames, while retaining generalization to different distributions.

## 3  Composable Part-Based Manipulation

In this work, our goal is to model functional actions involving an anchor object $\boldsymbol{A}$ that remains static and a function object $\boldsymbol{F}$ that is being actively manipulated. Shown in Fig. 2 (a), given a task $\boldsymbol{M}$ and the partial point clouds of two objects $\boldsymbol{X_A}$ and $\boldsymbol{X_F}$ in the world frame $\{\boldsymbol{W}\}$, we want to predict a sequence of $SE(3)$ transformations, i.e., $\mathcal{T}_{\boldsymbol{W}} = \{\boldsymbol{T_{W,1}}, .., \boldsymbol{T_{W,N}}\}$, which parameterized a trajectory of the function object $\boldsymbol{F}$ in the world frame in order to achieve the desired interaction with the anchor object $\boldsymbol{A}$ (e.g., pouring). Throughout the paper, we choose $N = 2$; i.e., we predict the starting pose and the ending poses of the object motion. Then, we use $SE(3)$ interpolation between the two poses to generate the continuous motion trajectory. We define that the object frames of $\{\boldsymbol{A}\}$ and $\{\boldsymbol{F}\}$ are centered at the centroids of the respective point clouds $\boldsymbol{X_A}$ and $\boldsymbol{X_F}$, and have the same orientation as the world frame *. Each transformation $\boldsymbol{T_W}$ in the world frame can thus be computed by the relative pose between the two objects $\boldsymbol{T_{AF}}$ as $\boldsymbol{T_W} = \boldsymbol{T_{WA}}\boldsymbol{T_{AF}}(\boldsymbol{T_{WF}})^{-1}$. A key challenge we aim to address is generalizing the functional actions from training objects to unseen object instances and, more importantly, novel object categories a robot may have never encountered during training.

---

*The transformation from $\{\boldsymbol{W}\}$ to an object frame can be computed given this definition. For example, $\boldsymbol{T_{WF}} = (\boldsymbol{R_{WF}}, \boldsymbol{t_{WF}})$, where $\boldsymbol{R_{WF}}$ is set to an identity matrix and $\boldsymbol{t_{WF}}$ is set to the centroid of $\boldsymbol{X_F}$.

## 3.1 Action as Part-Based Functional Correspondences

Composable part-based manipulation (CPM) models each action $M$ as a composition of functional correspondences between object parts. We formalize the symbolic representation of each correspondence $C \in \mathcal{C}_M$ as $\langle S_i, P_{A,j}, P_{F,k} \rangle$, where $\mathcal{C}_M$ is the set of correspondences for $M$, $S_i$ is a spatial relation, $P_{A,j}$ and $P_{F,k}$ are two parts of the anchor and the functional objects, respectively. Consider the example of pouring from a mug to a bowl, as depicted in Fig. 1. This "pour" action contains the following three correspondences: $\langle align, rim(\mathrm{mug}), rim(\mathrm{bowl}) \rangle$, $\langle tilt, body(\mathrm{mug}), body(\mathrm{bowl}) \rangle$, and $\langle facing\text{-}up, handle(\mathrm{mug}), body(\mathrm{bowl}) \rangle$.

The task of predicting robot motion can be cast as the task of finding a robot trajectory that simultaneously satisfies all the part-based functional correspondences. Instead of manually specifying these constraints given object point clouds and their poses, we propose to learn a neural network $g_\phi$ to recognize the functional parts of objects based on their point clouds and another learned generative model $p_\theta$ to parameterize a distribution of $\mathcal{T}$. Using $g_\phi$, we can extract point clouds for a given part, for example $g_\phi(X_F, P_{F,k}) = X_{P_{F,k}}$. Learning to recognize functional parts can be treated as predicting a per-point part segmentation problem and have been studied extensively in prior work [14, 15, 16, 27, 51]. Therefore, we focus on the second part which enables the robot to learn manipulation trajectories of objects, based on the recognized parts.

## 3.2 Generative Modeling of Functional Correspondences with Diffusion Models

For each functional correspondence tuple $\langle S_i, P_{A,j}, P_{F,k} \rangle$, we learn a generative distribution $p_{\theta, S_i}(\mathcal{T}_{P_{jk}} | X_{P_{A,j}}, X_{P_{F,k}})$. Here $\mathcal{T}_{P_{jk}}$ denotes the relative transformations $\mathcal{T}_{P_{A,j}P_{F,k}}^\dagger$. We use a point-cloud conditioned diffusion model to parameterize this distribution. In particular, each primitive diffusion denoise model $\epsilon_{\theta, S_i}$ takes in the current diffusion time step $t$, two part point clouds $X_{P_{A,j}}$ and $X_{P_{F,k}}$, and the noisy transformations $\mathcal{T}_{P_{jk}}$ as input, and predicts the noise over $\mathcal{T}_{P_{jk}}$. As illustrated in Fig. 2 (c), the model is based on a transformer encoder. First, we encode point clouds for the two parts separately using a point cloud transformer [52]. Then we encode each transformation using a trained MLP. We input the point cloud and transformation encodings, together with the diffusion time step $t$ to the transformer encoder. The output of the transformer encoder is the predicted noise over the transformations $\mathcal{T}_{P_{jk}}$. We provide details for the architecture in Appendix A.

During training, we optimize the following loss for randomly sampled diffusion time step $t$ and random Gaussian noise $\epsilon$ sampled from a multivariate Gaussian distribution:

$$\mathcal{L}_{\mathrm{MSE}} = \left\| \epsilon - \epsilon_{\theta, S_i} \left( \sqrt{1 - \beta_t} \mathcal{T}_{P_{jk}}^{(0)} + \sqrt{\beta_t} \epsilon \mid X_{P_{A,j}}, X_{P_{F,k}}, t \right) \right\|_2^2,$$

where $\mathcal{T}_{P_{jk}}^{(0)}$ is the target transformations to predict and $\beta_t$ is the diffusion noise schedule [53]. The added noise and the predicted noise are both in the tangent space of $SE(3)$. We build on the technique introduced for the $SE(3)$ Denoising Score Matching (DSM) model [40], but use Denoising Diffusion Probabilistic Model (DDPM) [53] for more stable training. In practice, we first compute the exponential map of the transformations and then apply the noise. This can be viewed as predicting the score function for an exponential energy function of $SE(3)$ poses.

## 3.3 Inference-Time Composition of Diffusion Models

One of the key features of diffusion models is their compositionality. That is, suppose we have a set of diffusion models, each trained for one specific type of functional correspondences, we can combine their predicted noises during inference time to generate a trajectory that adheres to all functional correspondences, as illustrated in Fig. 2 (b). Since each diffusion model implicitly parameterizes an energy-based model: $p_{\theta, S_i}(\mathcal{T}|\cdot) \propto \exp(-E_{\theta, S_i}(\mathcal{T}|\cdot))$ through its noise prediction [48, 49], sampling from the composition of the diffusion models corresponds to sampling from the "intersection" of distributions for the individual functional correspondences, or formally, from $\prod_{C \in \mathcal{C}_M} p_{\theta, S_i}(\mathcal{T}|\cdot)$.

---

$^\dagger$Similar to the definition of the object frame, the part frames $\{P_{A,j}\}$ and $\{P_{F,k}\}$ are centered at the centroids of the respective point clouds $X_{P_{A,j}}$ and $X_{P_{F,k}}$ and have the same orientation as the world frame.

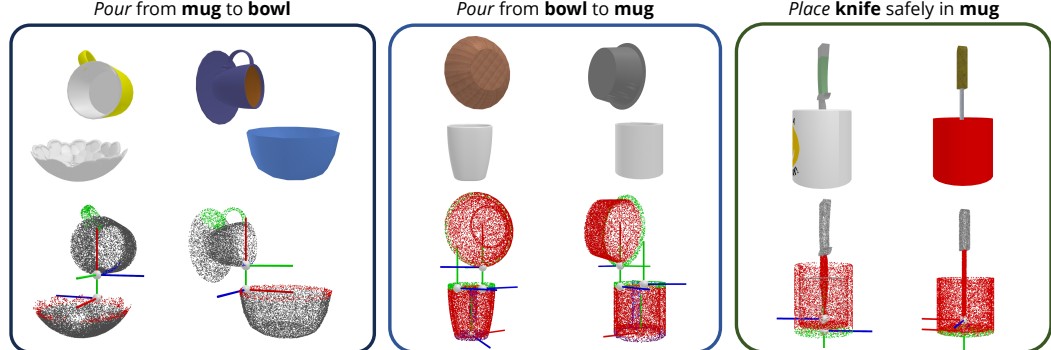

**Figure 3:** We generate task demonstrations using the PartNet and ShapeNetSem datasets for the "pouring" and "safe placing" tasks. We create demonstrations for a variety of function and anchor object combinations.

In particular, during inference time, starting from $\mathcal{T}_{AF}^{(T)}$ randomly sampled from standard Gaussian distributions, given the set of constraints $\mathcal{C}_M$, we iteratively update the pose prediction by:

$$\mathcal{T}_{AF}^{(t-1)} = \frac{1}{\alpha_t} \left( \mathcal{T}_{AF}^{(t)} - \frac{1 - \alpha_t}{\sqrt{1 - \bar{\alpha}_t}} \sum_{C \in \mathcal{C}_M} \epsilon_{\theta, S_i} \left( f_{\texttt{to-part}}(\mathcal{T}_{AF}^{(t)}) \mid X_{P_{A,j}}, X_{P_{F,k}}, t \right) \right) + \sigma_t \epsilon,$$

where $T$ is the number of diffusion steps, $\alpha_t = 1 - \beta_t$ is the denoise schedule, $\bar{\alpha}_t = \prod_{i=1}^{T} \alpha_t$ is the cumulated denoise schedule, $\sigma_t$ is a fixed sampling-time noise schedule, and $\epsilon$ is a randomly sampled Gaussian noise. The differentiable operation $f_{\texttt{to-part}}$ takes $\mathcal{T}_{AF}^{(t)}$ and transforms it to the part frame $P_{jk}$ by $(\mathcal{T}_{AP_{A,j}})^{-1}\mathcal{T}_{AF}^{(t)}\mathcal{T}_{FP_{F,k}}$, for which each individual diffusion model is trained on.

## 4 Data Collection

We demonstrate CPM on the "pouring" and "safe placing" tasks. These two tasks require different functional affordances. The pouring action pours from an anchor object to a target object, and requires alignment of *rims*, collision avoidance of the *handle* and the container *body*, and *body* tilt. The safe-placing action places a sharp function object into an anchor object, and requires *head* containment for safety, *tip* touching *bottom*, and a *body-body* placement constraint. To validate our approach, we collect 4522 successful demonstrations for pouring and 2836 successful demonstrations for safe placing. To generate the demonstrations, we first source 13 categories of 3D objects from PartNet [54] and the subset of ShapeNetSem [55] objects categorized in the Acronym dataset [56]. We then extract aligned parts either from segmentations and category-level canonical poses in PartNet or from manually labeled 3D keypoints for ShapeNetSem objects. We procedurally generate parameters of the actions from the aligned parts (as illustrated in Fig. 3), simulate the interactions by tracing the trajectories defined by the parameters, and render RGB-D images using multiple cameres set up in the simulator. Details of the dataset are presented in Appendix C.

## 5 Experiments

The subsequent section will showcase the performance of CPM in comparison to baselines and other variants of our method in simulation. In particular, we evaluate in two important generalization settings: 1) generalization to novel object instances from seen object categories, and 2) generalization to object instances from *unseen* object categories. We then discuss the deployment of CPM trained in simulation on a real robot.

### 5.1 Experimental Setup

We evaluate all methods in the PyBullet physics simulator [57]. To isolate the problem of predicting target transformations $\mathcal{T}$ from other components of the system (e.g., grasp sampling and motion planning), we actuate the center of mass of the function object $F$. We report average task completion scores from 1500 trials indicating failure (0) and success (100), with credits assigned for partial completion. The score is computed based on model-based classifiers designed for each task. To test

**Table 1:** CPM demonstrates strong generalization to novel instances of objects within seen categories.

| Model | Pouring | Safe Placing |
|---|---|---|
| Transformer-BC | 19.21 | 37.11 |
| TAX-Pose | 21.71 | **76.97** |
| PC-DDPM | 75.83 | 51.55 |
| Part-Aware PC-DDPM | 75.28 | 42.68 |
| CPM (ours) | **80.00** | 70.99 |

**Table 2:** CPM demonstrates strong generalization to function objects from unseen object categories.

| Model | Pouring | | | | Safe Placing | | |
|---|---|---|---|---|---|---|---|
| | Bowl | Glass | Mug | Pan | Fork | Pen | Scissors |
| Transformer-BC | 10.23 | 20.91 | 6.06 | 32.04 | 26.15 | 31.93 | 26.44 |
| TAX-Pose | 23.32 | 3.82 | 8.64 | 46.14 | 50.90 | **67.60** | 36.80 |
| PC-DDPM | 63.02 | 75.95 | 71.39 | 64.39 | 40.60 | 46.63 | 32.34 |
| Part-Aware PC-DDPM | 58.98 | 72.11 | 67.11 | **66.17** | 39.76 | 48.04 | 28.15 |
| CPM (ours) | **79.32** | **81.44** | **77.57** | 62.13 | **55.94** | 59.45 | **63.35** |

generalization to novel objects from seen categories, we randomly split the data for each task $M$ into 80% training and 20% testing. To test generalization to unseen object categories, we conduct a separate experiment for each target category of the function objects, where we withhold data involving the target category and train on the remaining data. Details of the evaluation are discussed in Appendix D. We present results with binary success as metric in Appendix E.

## 5.2 Compared Methods

**Baselines.** We compare CPM with four main baselines. The first is Transformer-BC, which uses a multimodal transformer encoder-decoder from prior work [30] to condition on point clouds of the objects and autoregressively predict target transformations. The second baseline is based on TAX-Pose [37] which predicts relative poses between two objects from point-wise soft correspondences. The third is PC-DDPM; similar to recent work [40, 47], a conditional denoising diffusion probabilistic model [53] is trained to predict target transformations based on input point clouds of both the function and the anchor objects. The fourth baseline is the Part-Aware PC-DDPM, which takes in both point clouds of the objects and per-point segmentation masks that indicate object parts. We discuss the baseline implementations in details in Appendix B.

**CPM variants.** We evaluate several variants of our model. The first is DDPM with 6D rotation representation instead of $SE(3)$. This variant of CPM learns different diffusion models for different parts. However, it does not compose pose predictions in different part frames. This model is directly adapted from existing composable diffusion models for image generation [48, 49]. The second is DDPM with training-time composition; this model jointly train all primitive diffusion models by composes thier noise predictions at training time. The last group are the individual primitive diffusion models, which use single DDPM models corresponding to different part-part correspondences, without any composition.

## 5.3 Simulation Results

**Comparisons to baselines.** We evaluate CPM's generalization capability in two settings. First, Table 1 shows a comparison of generalization to novel objects from seen categories. Overall, our model achieves strong performance on both tasks of "pouring" and "safe placing". We note that TAX-Pose struggles with pouring that requires modeling multimodal actions because the method extracts a single relative pose estimate from a fixed set of correspondences. The autoregressive Transformer-BC is also not enough to capture the full distribution of the pouring action. We note that although Part-Aware PC-DDPM leverages the same part segmentation as CPM, it fails to achieve stronger performance compared to the PC-DDPM baseline, which only uses the object point clouds as input. We attribute this to its potential overfitting to the part segmentations within the training data. By contrast, CPM is able to effectively leverage part segmentations by learning primitive diffusion models and composing them at inference time. Our model shows substantial improvements in the

**Table 3:** We ablate the contributions of CPM on the ability to generalize to novel categories of objects.

| Target Pose Rep | Part Frames | Composition | Pouring | Safe Placing |
|---|---|---|---|---|
| 6D Rot + 3D Trans | No | Inf-time | 71.22 | **68.77** |
| $SE(3)$ | Yes | Train-time | 69.89 | 48.46 |
| $SE(3)$ | Yes | Inf-time | **75.11** | 59.58 |

**Table 4:** We explore the effect of composition, comparing to individual diffusion models, in generalization across both "pouring" and "safe placing" tasks. *We note that for the *align* and *facing-up* evaluation, a small percentage of examples were removed as they do not contain the involved parts in the partial object point clouds.

| Pouring | | Safe Placing | |
|---|---|---|---|
| $\langle align, rim, rim \rangle$ | 70.05* | $\langle contain, head, body \rangle$ | 41.22 |
| $\langle facing\text{-}up, handle, body \rangle$ | 16.42* | $\langle touch, tip, bottom \rangle$ | 9.34 |
| $\langle tilt, body, body \rangle$ | 68.69 | $\langle place, body, body \rangle$ | 39.86 |
| CPM | **75.11** | CPM | **59.58** |

"safe placing" task compared to other diffusion-based methods, largely due to each part constraint significantly restricting the target pose distribution in this task. For instance, the constraint that requires the *tip* of the function object to touch the *bottom* of the anchor object effectively constrains the target pose.

Our second set of experiments assesses the model's capacity to generalize to *unseen* object categories, thereby highlighting the efficacy of part-based correspondences. Results can be found in Table 2. Remarkably, CPM demonstrates its capability to generalize across object categories for both tasks in a zero-shot manner. CPM's performance dips slightly for pans as the *rim*'s of pans are significantly larger compared to *rim*'s encountered during training (for example, those of bowls and mugs). As a comparison, all baselines fall short in consistently generalizing to new categories for both tasks. TAX-Pose is not able to maintain strong performance for safe placing when generalizing to more geometrically complicated objects including scissors and forks. Our methods are robust to changes in local geometry and overall topology by leveraging compositions of part-based correspondences.

**Ablation.** First, we assess the significance of our $SE(3)$ encoding, part frame-based transformation, and inference-time composition within the context of generalizing to unseen categories of objects. As depicted in Table 3, our full CPM with part frames and inference-time composition shows superior performance compared to the model trained with training-time composition. This verifies the importance of our designs to support part-based composition and generalization. Compared to the variant based on 6D Rotation + 3D Translation encoding, CPM yields a better performance on the pouring task, a scenario where the rotation of the function object plays a pivotal role. On the safe placing task, which involves less rotation of objects, we observe a more comparable performance with our model. These results highlight the importance of $SE(3)$ diffusion model in rotation prediction.

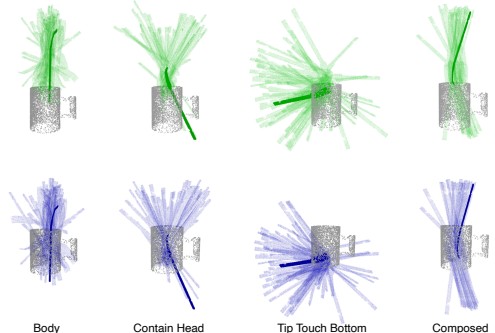

Body | Contain Head | Tip Touch Bottom | Composed

**Figure 4:** We illustrate the learned distribution of each primitive diffusion model, which generates diverse samples conforming to the specified constraints, as well as the distribution from the combined full CPM model. The highest-ranked sample is highlighted.

Second, we compare the performance of composed part-based diffusion models with the performance of primitive diffusion models. Shown in Table 4, the composed model outperforms individual diffusion models, showing the efficacy of our composition paradigm. In addition, these results show the importance of different part-based constraints for the given tasks. In the "pouring" task, *align* and *tilt* strongly constrain the target pose for the function object, while for the "safe placing" task, *contain* and *place* constraints are more salient. Fig. 4 provides a qualitative visualization by showcasing the part-conditioned distribution associated with each individual diffusion model for

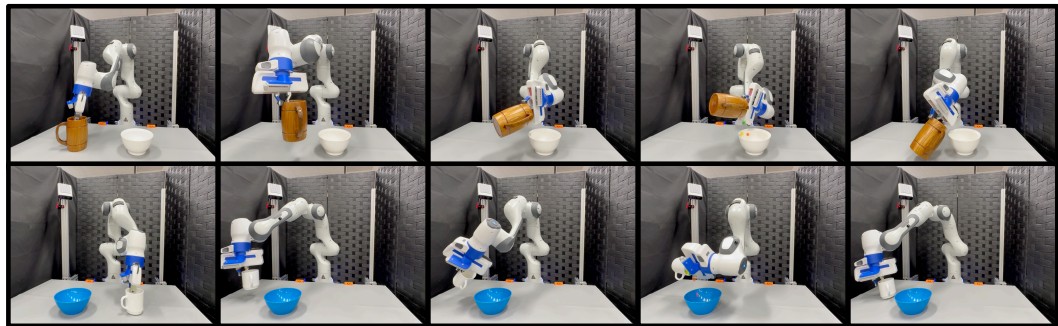

**Figure 5:** We show sampled frames from trajectories of CPM's policy. The model is trained only on demonstrations with pans, bowls, and wine glasses in simulation and generalizes to mugs in the real world.

various constraints, as well as the corresponding composed distribution. The quantitative performance of *contain* and *place* primitive models for these tasks aligns with this qualitative comparison, as they have learned distributions that are close to the composed model. The CPM paradigm allows us to train each primitive diffusion model independently, encouraging each model to concentrate on distinct functional affordances, thus enabling them to learn and generalize to diverse distributions of samples. During inference, the composition of distributions learned by individual models enables CPM to find solutions that satisfy all correspondence constraints.

### 5.4 Real-World Transfer

Finally, we show a real-world robot manipulation experiment for the "pouring" task, highlighting the transferability of our CPM to real-world manipulation. In this setting, we use the primitive diffusion models trained on simulation data with function objects of glasses, pans, and bowls, and zero-shot transfer to mugs in the real-world experiment. Our setup includes a Franka Emika robot mounted in a tabletop environment. To conduct pouring, we perform plane segmentation and k-means clustering to extract object point clouds from the scene point cloud captured by two calibrated Azure Kinect RGB-D cameras. Next, we apply a pre-trained point transformer (PT) model [58] for part segmentation. The segmentation model is trained on simulation data only. We then apply CPM trained in simulation for the pouring task. To execute the trajectory, we use the Contact-GraspNet [59] to sample robot grasps on the function object and Operational Space Controller [60] with impedance from Deoxys [60] to following a sequence of end-effector pose waypoints computed from the target transformations. Figure 5 shows our real-world setup and example trajectories predicted by CPM on unseen mugs with different shapes and sizes.

## 6 Limitations and Conclusion

We introduced composable part-based manipulation (CPM), as an approach that leverages object-part decomposition and part-part correspondences for robotic manipulation. We show that representing actions as combinations of constraints between object parts enables strong generalization. Through the composition of primitive diffusion models, we gain generalization capabilities across novel instances of objects as well as unseen object categories, in simulation and in real-world robot experiments.

In this paper, we focus on manipulation tasks involving two objects. Extending CPM to learn skills involving more objects would be important for future work, in particular for manipulating piles or stacks of objects. Second, we parameterize each manipulation action by the starting and ending poses. Extending the transformer-based diffusion model to output more waypoints to parameterize longer trajectory is important for potentially a wider range of tasks. In addition, CPM does not model temporal constraints over the trajectory. One possible extension is to learn trajectory samplers for temporal constraints and trajectories with loops. CPM assumes external part segmentations. Although many categories can be segmented by off-the-shelf computer vision models [26], extending the system to jointly learn or finetune part segmentation is important. Finally, composing a larger number of diffusion models may require more efficient sampling techniques such as [61]. We provide an extended discussion of CPM's assumptions in Appendix F and suggest directions for future research.

**Acknowledgments**

We extend our gratitude to the members of the NVIDIA Seattle Robotics Lab, the RAIL research lab at Georgia Tech, and the Stanford Vision and Learning Lab for insightful discussions. This work is in part supported by NSF grant 2214177, 2211258, AFOSR grant FA9550-22-1-0249, FA9550-23-1-0127, ONR MURI grant N00014-22-1-2740, the Stanford Institute for Human-Centered Artificial Intelligence (HAI), the MIT-IBM Watson Lab, the MIT Quest for Intelligence, the Center for Brain, Minds, and Machines (CBMM, funded by NSF STC award CCF-1231216), and Analog Devices, JPMC, and Salesforce. Any opinions, findings, and conclusions or recommendations expressed in this material are those of the authors and do not necessarily reflect the views of our sponsors.

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

# A Network Architecture

For each functional correspondence $\langle \boldsymbol{S}_i, \boldsymbol{P}_{\boldsymbol{A},j}, \boldsymbol{P}_{\boldsymbol{F},k} \rangle$, we aim to learn a generative distribution $p_{\theta,\boldsymbol{S}_i}(\mathcal{T}_{\boldsymbol{P}_{jk}} | \boldsymbol{X}_{\boldsymbol{P}_{\boldsymbol{A},j}}, \boldsymbol{X}_{\boldsymbol{P}_{\boldsymbol{F},k}})$. Here we discuss the network architecture for primitive diffusion model $\epsilon_{\theta,\boldsymbol{S}_i}$ that learns to estimate the generative distribution. We leverage modality-specific encoders to convert the multimodal inputs to latent tokens that are later processed by a transformer network.

**Object encoder.** Given part point clouds $\boldsymbol{X}_{\boldsymbol{P}_{\boldsymbol{A},j}}$ and $\boldsymbol{X}_{\boldsymbol{P}_{\boldsymbol{F},k}}$, we use a learned encoder $h_p$ to encode each part separately as $h_p(\boldsymbol{X}_{\boldsymbol{P}_{\boldsymbol{A},j}})$ and $h_p(\boldsymbol{X}_{\boldsymbol{P}_{\boldsymbol{F},k}})$. This encoder is built on the Point Cloud Transformer (PCT) [52].

**Diffusion encodings.** Since the goal transformations $\mathcal{T}_{\boldsymbol{P}_{jk}} = \{\boldsymbol{T}_{\boldsymbol{P}_{jk},n}\}_{n=1}^N$ are iteratively refined by the diffusion model and need to feed back to the model during inference, we use a MLP to encode each goal transformation separately $h_T(\boldsymbol{T}_{\boldsymbol{P}_{jk},n})$. To compute the time-dependent Gaussian posterior for reverse diffusion, we obtain a latent code for $t$ using a Sinusoidal embedding $h_{time}(t)$.

**Positional encoding.** We use a learned position embedding $h_{pos}(l)$ to indicate the position index $l$ of the part point clouds and poses in input sequences to the subsequent transformer.

**Diffusion Transformer.** The diffusion model predicts the goal poses $\mathcal{T}_{\boldsymbol{P}_{jk}}^{(0)}$ starting from the last time step of the reverse diffusion process $\mathcal{T}_{\boldsymbol{P}_{jk}}^{(T)} \sim \mathcal{N}(0, \mathcal{I})$, which is sampled from a multivariate normal distribution with independent components. We use a transformer encoder as the backbone for the diffusion model $\epsilon_{\theta,\boldsymbol{S}_i}\left(\{\boldsymbol{T}_{\boldsymbol{P}_{jk},n}^{(t)}\}_{n=1}^N \mid \boldsymbol{X}_{\boldsymbol{P}_{\boldsymbol{A},j}}, \boldsymbol{X}_{\boldsymbol{P}_{\boldsymbol{F},k}}, t\right)$, which predicts the time-dependent noise $\{\epsilon_1^{(t)}, ..., \epsilon_N^{(t)}\}$. We obtain the transformer input for the parts $\chi$ and the target poses $\tau$ as

$$\chi_{\boldsymbol{A}}^{(t)} = [h_p(\boldsymbol{X}_{\boldsymbol{P}_{\boldsymbol{A},j}}); h_{pos}(0); h_{time}(t)]$$
$$\chi_{\boldsymbol{F}}^{(t)} = [h_p(\boldsymbol{X}_{\boldsymbol{P}_{\boldsymbol{F},k}}); h_{pos}(1); h_{time}(t)]$$
$$\tau_n^{(t)} = [h_T(\boldsymbol{T}_{\boldsymbol{P}_{jk},n}^{(t)}); h_{pos}(n-2); h_{time}(t)]$$

where $[;]$ is the concatenation at the feature dimension. The model takes in the sequence $\{\chi_{\boldsymbol{A}}^{(t)}, \chi_{\boldsymbol{F}}^{(t)}, \tau_1^{(t)}, ..., \tau_N^{(t)}\}$ and predicts $\{\epsilon_1^{(t)}, ..., \epsilon_N^{(t)}\}$ for the object poses.

**Parameters.** We provide network and training parameters in Table A1.

**Table A1:** Model Parameters

| Parameter | Value |
|---|---|
| Number of $\boldsymbol{P}_{\boldsymbol{A},j}$ and $\boldsymbol{P}_{\boldsymbol{F},k}$ points | 512 |
| PCT point cloud encoder $h_p$ out dim | 200 |
| Position embedding $h_{pos}$ | learned embedding |
| Position embedding $h_{pos}$ dim | 16 |
| Time embedding $h_{time}$ | Sinusoidal |
| Time embedding $h_{time}$ dim | 40 |
| Pose encoder $h_T$ out dim | 200 |
| Transformer number of layers | 4 |
| Transformer number of heads | 4 |
| Transformer hidden dim | 128 |
| Transformer dropout | 0.0 |
| Diffusion steps $T$ | 200 |
| Diffusion noise schedule $\beta_t$ | Linear |
| Start value $\beta_0$ | 0.0001 |
| End value $\beta_T$ | 0.02 |
| Loss | Huber |
| Epochs | 2000 |
| Optimizer | Adam |
| Learning rate | 1e-4 |
| Gradient clip value | 1.0 |

# B Implementation Details for Baselines

We discuss the implementation of each baseline below:

- Transformer-BC: this baseline takes point clouds $X_A$ and $X_F$ as input, and predicts target transformations $\mathcal{T}_{AF}$. This baseline uses a multimodal transformer encoder-decoder from prior work [30] to condition on point clouds of the objects and autoregressively predict target transformations. The point clouds are first individually encoded with a point cloud transformer [52]. The point cloud embeddings are fed to the transformer encoder. The transformer decoder autoregressively decodes the target poses $\{T_{AF,1}, .., T_{AF,N}\}$.

- TAX-Pose: this baseline takes point clouds $X_A$ and $X_F$ as input, and predicts target transformations $\mathcal{T}_{AF}$. We use the code and hyperparameters from the official repository ‡. We use the variant that does not require pretrained object embeddings because we use different objects from the paper. As discussed in Appendix F.1.2 of the original paper, pretraining mainly helps to reduce training time. Because the TAX-Pose model only predicts one relative pose for each pair of point clouds, we learn a separate model for each transformation in $\mathcal{T}_{AF}$. Specifically, one TAX-Pose is trained to predict start pose and another TAX-Pose is trained to predict end pose.

- PC-DDPM: this baseline takes point clouds $X_A$ and $X_F$ as input, and predicts target transformations $\mathcal{T}_{AF}$. Similar to recent work [40, 47], a conditional denoising diffusion probabilistic model [53] is trained to predict target transformations based on input point clouds of both the function and the anchor objects. This model has the same architecture, including encoders, latent embeddings, and the diffusion transformer, as the primitive diffusion models, which is discussed in Appendix A.

- Part-Aware PC-DDPM: this baseline takes point clouds $X_A \in \mathbb{R}^{N_X \times 3}$ and $X_F \in \mathbb{R}^{N_X \times 3}$ and two segmentation masks $I_A \in \mathbb{R}^{N_X \times N_I}$ and $I_F \in \mathbb{R}^{N_X \times N_I}$ as input, and predicts target transformations $\mathcal{T}_{AF}$. $N_X$ is the number of points for each point cloud and $N_I$ is the number of known object parts. Each channel of the segmentation mask is a binary mask indicating points for a specific part. Each segmentation mask encodes all parts that can be extracted from an object point cloud. For simulation experiment, the segmentation masks come from groundtruth part segmentation. While CPM use the segmentation masks to extract part point clouds, this baseline directly encode the segmentation mask together with the object point cloud. This baseline shares most of the network architecture as PC-DDPM except that point cloud encoder now encodes $[X_A; I_A] \in \mathbb{R}^{N_X \times (3+N_I)}$ and $[X_F; I_F] \in \mathbb{R}^{N_X \times (3+N_I)}$.

# C Dataset Details

In total, we collected 4522 successful demonstrations for pouring and 2836 successful demonstrations for safe placing. For each experiment, we use a subset of these demonstrations for training the models, and the remaining data for initializing the simulation. We provide a breakdown of the dataset in Table A2. Because the expert policies do not have 100% success rate, the models will only be trained on the successful demonstrations. Below we discuss our data collection process in details.

**Sourcing 3D objects.** We source a wide variety of 3D objects from PartNet [54] and the subset of ShapeNetSem [55] objects categorized in the Acronym dataset [56]. We use 13 object categories to investigate generalization, including *mug*, *pan*, *bowl*, *wine glass*, *knife*, *can opener*, *scissors*, *screwdriver*, *fork*, *spoon*, *marker*, *pen*, and *flashlight*. Some object categories are reused for different tasks; for example, *mug* is used as an anchor for safe placing but also as an object for pouring.

**Extracting aligned parts.** Our generative diffusion models uses part segmentations of objects to learn primitive diffusion models. For 3D objects from PartNet, we use the segmentations provided in

---

‡Code from https://github.com/r-pad/taxpose.

**Table A2:** Simulation and Demonstration Data

| Task | Object | Source | Number of Simulations | Number of Success Demonstrations |
|------|--------|--------|----------------------|----------------------------------|
| Safe Placing | Pen | PartNet | 1000 | 568 |
| | Fork | PartNet | 1000 | 390 |
| | ScrewDriver | PartNet | 1000 | 145 |
| | Spoon | PartNet | 1000 | 410 |
| | Knife | Acronym | 1000 | 496 |
| | Scissors | PartNet | 1000 | 354 |
| | Flashlight | PartNet | 1000 | 141 |
| | CanOpener | PartNet | 1000 | 101 |
| | Marker | PartNet | 1000 | 231 |
| Pouring | Mug | PartNet | 2000 | 1051 |
| | WineGlass | PartNet | 2000 | 1542 |
| | Bowl | Acronym | 2000 | 776 |
| | Pan | PartNet | 2000 | 1153 |

the dataset. For 3D objects from ShapeNetSem, we first label 3D keypoints, then from the labeled keypoints, we procedurally extract parts. As ShapeNet provides canonical poses for 3D models, we can also align the extracted functional parts for each object category.

**Simulating trajectories and rendering.** We simulate the robot-object interactions by tracing the trajectories defined by the parameters. We first use multiple cameras to render RGB-D images, which yield realistic object point clouds. We then map the functional parts to the point clouds with the correct transformation and scaling. Finally, we obtain point cloud segments of each affordance part. Because these parts are extracted from the rendered point clouds, they can be incomplete, which increases the robustness of our method and helps transferability to real-world settings.

# D   Evaluation Details

In Section 5, we report task completion scores. For each experiment, we randomly draw 100 samples from the withheld testing data to initialize simulation for evaluation. This procedure ensures that the action can be successfully performed for the pair of anchor and function objects. To systematically evaluate multimodal actions (e.g, pouring from different directions), we sample from each model 5 times and simulate the predicted actions. We repeat each experiment with 3 different random seeds, resulting in a total of 1500 trials.

The task score indicates task completion between failure (0) and success (100), with credits assigned for partial completion. The score is computed based on model-based classifiers designed for each task. Now we describe how the score is computed in more detail:

- Pouring: we first use PyBullet's collision test to check whether the function object and anchor object will ever interpenetrate during the execution of the action by rigidly transforming the function object to the predicted poses. If the objects interpenetrate, we assign a score of zero because the action cannot be realistically executed. Then we simulate the pouring action, and use the percentage of particles successfully transferred from the function object to the anchor object as the partial score.

- Safe Placing: similar to pouring, we check interpenetration for the start pose of the placement action. If the objects interpenetrate, we assign a score of zero. Then we simulate the placement action until contact between the anchor and function object. If the orientation of the function object is incorrect (e.g., the blade of the knife is outside of the container), we assign a score of zero. If the orientation is correct, the percentage of the trajectory parameterized by the predicted transformations that is successfully executed is used as the partial score.

# E  Additional Results

Besides reporting the task completion scores, we include additional task success rates in Table A3 and Table A4. For pouring, a trial is considered successful if there is no interpenetration between objects and 70% of particles are successfully transferred. For safe placing, a successful trial requires no interpenetration at the predicted start pose for the function object, correct orientation of the function object, and 70% of the predicting trajectory being successfully executed without collision between objects. We observe similar trends as the results presented in Section 5.

**Table A3:** CPM shows strong generalization to novel instances of objects within seen categories.

| Model | Pouring | Safe Placing |
|---|---|---|
| Transformer-BC | 17.53±3.13 | 33.27±2.14 |
| TAX-Pose | 21.33±0.58 | **74.00±1.00** |
| PC-DDPM | 70.67±1.27 | 48.73±2.97 |
| Part-Aware PC-DDPM | 73.60±2.60 | 36.53±2.20 |
| CPM (ours) | **76.87±1.70** | 68.87±2.25 |

**Table A4:** CPM demonstrates strong generalization to function objects from unseen object categories.

| Model | Pouring | | | | Safe Placing | | |
|---|---|---|---|---|---|---|---|
| | Bowl | Glass | Mug | Pan | Fork | Pen | Scissors |
| Transformer-BC | 10.00±2.51 | 19.20±0.92 | 5.80±1.93 | 29.33±2.04 | 24.00±1.51 | 27.33±2.34 | 18.47±2.93 |
| TAX-Pose | 21.00±1.00 | 3.00±1.00 | 8.00±1.00 | 42.67±2.08 | 47.67±3.21 | **62.67±4.04** | 33.33±1.15 |
| PC-DDPM | 56.53±2.00 | 70.67±3.06 | 68.67±4.31 | 59.93±2.80 | 38.00±3.83 | 43.47±1.68 | 28.47±0.83 |
| Part-Aware PC-DDPM | 54.87±2.10 | 68.33±2.97 | 65.20±4.61 | **62.00±3.56** | 28.67±1.68 | 42.40±3.12 | 17.67±2.70 |
| CPM (ours) | **76.40±1.78** | **78.93±3.14** | **76.00±5.26** | 54.67±1.50 | **53.93±2.91** | 56.53±2.04 | **62.07±1.72** |

# F  Assumptions

During training, our method assumes 1) a description of the manipulation skill as a set of part correspondences, 2) access to the dataset of successful trajectories, and 3) access to part segmentations for objects in the dataset. During testing, our method assumes the part segmentations for objects being manipulated. We contend that these assumptions align with our current focus. Nonetheless, subsequent research should aim to address them.

First, the description of manipulation skills is in symbolic text, e.g., pouring from mugs to bowls contains three constraints. They can be easily annotated by humans as there is no need to specify any continuous parameters or mathematical formulas. An interesting future direction is to leverage large language models to more efficiently extract constraints. CPM then learns the grounding of these constraints from data.

Second, we assume access to successful manipulation trajectories. That is, we do not assume any additional annotations, such as programs for generating these trajectories. The key focus of the paper is to improve the data efficiency of learning such skills, in particular for generalization across categories. An important future direction is to improve the data efficiency of this method and learn from noisy human demonstrations.

Finally, relying on external part segmentation is limiting, but 2D or 3D part segmentation models are generally available for many object categories [15, 16, 26]. An exciting future direction is to extend the current framework to automatically discover functional part segmentations leveraging manipulation data.

