# OpenReview forum: "Composable Part-Based Manipulation"
_robot-learning.org/CoRL/2023/Conference — CoRL 2023 Poster_

### Official Review · Reviewer_3XuZ · 2023-07-05

**Confidence:** 5
**Originality:** Very Good
**Technical Quality:** Very Good
**Clarity Of Presentation:** Very Good
**Impact:** 3

**Recommendation:**

Weak Accept: I recommend accepting the paper, but will not argue for my recommendation if the majority of other reviewers have a different opinion.

**Review:**


## Strengths

The method leverages the inductive bias that parts and their language descriptions capture their functional affordances, formalizing an action as a set of rules relating to object parts that can be learned. This approach could be generalized to novel unseen instances without the need for massive datasets.

The related work is well structured and sufficient.

Part-based methods allow easy generalization to objects with slightly different part structures (such as cups to mugs).

An interesting novel application of diffusion models to robotics.

The experimental results are convincing of the method's usefulness.

## Weaknesses

The method assumes access to a set of symbolic constraints associated with each task.
Additionally, it is unclear how constraints, which are central for this method, are defined - are they specified by a user with specific parameters optimized by the model, are they fully pre-defined, are they fully learned?

Only two tasks are considered for generalization to unseen categories, of which the second is not particularly impressive. Other tasks that could have been considered include handled object on rack, gripper on various objects.

It is unclear why the method was not compared with other baselines such as TAX pose and NDF which tackle a similar problem

The limitations are only briefly discussed.

## Other minor comments

I suggest rephrasing the research contribution of the work. A research contribution is a novel and significant addition to the field of study that advances the existing knowledge, theories or practices.

Modelling manipulation based on part-part correspondences is not novel – many works within robot affordance learning research have done this (e.g. refer to the works of Eren Aksoy, or some of the references cited). Additionally, the third contribution is just a result of the other two.

Some citations are capitalized wrong.

L73: should this be “within” instead of “between”

Figure 2 and its caption are unclear – How does the combined noise feed back into the sampling?

Where is the point cloud transformer in the figure (referred to on L138)? It has no clear start or end, and the images used to represent pose are confusing.

L145 – T0 is different from the T0 used in the equation.


**Quality Of The Limitations Section:**

Additional details required

**Questions For Rebuttal:**

How are the constraints defined in the context of this work?

L166 ``’the composition corresponding to maximize’ - what does this mean?

What tasks were considered for testing this method, were there any that were not considered due to limitations of this particular model?

L199 claims that rendered point clouds can be noisy – was artificial noise applied to the point clouds? Typically, simulated point clouds are as ``perfect’’ as possible.

L185 – how are the parts procedurally extracted?

While it is not solving the problem in the same way, is it not appropriate to compare the method with approaches such as TAX-Pose and R-/NDF? Why/Why not?


**Robotics Focus:**

Sufficient demonstration on hardware

**Summary Of Paper:**

The paper proposes composable part-based simulation – it’s tackling the problem of learning and genralizing robot manipulation skills through part-part correspondences. The method proposes the use of composable diffusion models which capture the various inter-object relationships.


**Summary Of Recommendation:**

Overall, the method is interesting and appears useful, but there are some unanswered questions. I am recommending a weak acceptance as I believe the concerns are all addressable, but I can ultimately be convinced either way depending on the author’s responses and the comments of other reviewers.

---

### Official Review · Reviewer_ayCc · 2023-07-13

**Confidence:** 4
**Originality:** Very Good
**Technical Quality:** Good
**Clarity Of Presentation:** Fair
**Impact:** 4

**Recommendation:**

Strong Reject: I recommend rejecting the paper and will argue for my recommendation even if other reviewers hold a different opinion.

**Review:**

Strengths
- Authors show real-world demonstration showing impressive successful sim-to-real transfer, including generalization to unseen mugs
- The authors proposed method of using part-part correspondences as the basis of modelling manipulation actions, and the integration of this with diffusion models is very interesting and novel. I believe this paper is very relevant to the CoRL community.

Weaknesses
The main weakness of the paper is that in places it is unclear and missing key details. In addition I believe there is room for improvement in the baselines used for comparison.

- On line 135, and in the equation below line 144, the authors state that each individual takes as input two object point-clouds, but in the equation below line 160, these are now partial point-clouds. From the rest of the text, particularly the discussions of the baselines, I am inclined to think that the authors use partial point-clouds of the parts as input but it is unclear.
- In the section 'Inference-time combination' the authors discuss performing the diffusion in SE3. This is further ablated in the experiments with the model with a 6D rotation parameterization. However, performing diffusion on SE3 is not a contribution of this paper, since that has been proposed in [40]. The section 'Inference-time combination' should be rewritten to make clear that this is from prior work and not part of the authors contribution.
- I found the description of the baselines and their relationship to CPM very unclear. In particular, for both PC-DDPM and part-aware-PC-DDPM, it is very unclear to me what their input/output is. Is there a model per correspondence as with CPM, or a model per task? If neither, how are the correspondences/tasks encoded? This becomes especially important for part-aware-PC-DDPM because from the text I do not understand what the difference between part-aware-PC-DDPM and CPM is.
- I cannot find anywhere in the text a definition of the performance metric. Is it a success rate? Over how many evaluations? The authors proposed approach shows improved performance including on unseen tasks according to this metric, but I cannot list this as a strength of the paper until I know what this performance metric means.
- The authors baselines are only other diffusion models, and all of the baselines are developed by the authors. I think they would be much more convincing if they were contrasted from another method from the literature. Perhaps something from the offline RL literature such as CQL.
- One key limitation that is not discussed in the limitations section is the quite restrictive set of tasks that this method can be applied to, namely consisting of two-object interactions with only one object actively manipulated.

**Quality Of The Limitations Section:**

Additional details required

**Questions For Rebuttal:**

- Revise the manuscript to improve clarity (from above section)
- Incorporate additional non-diffusion baseline

**Robotics Focus:**

Sufficient demonstration on hardware

**Summary Of Paper:**

This paper proposes an approach for learning manipulation skills which involve the interaction of two objects, with one object actively controlled by the robot. The key idea is to decompose objects into their constituent parts, and define skills in terms of the correspondences between parts. The authors propose learning separate diffusion models per correspondence, which generates initial and final poses for the actively controlled object conditioned on point-clouds. Skills involving multiple correspondences are then defined by the composition of the appropriate diffusion models. The authors demonstrate their approach on two tasks and show that the method can achieve generalization to unseen object categories. The authors also include a real-world demonstration of the method.

**Summary Of Recommendation:**

I think this paper shows promise, but due to it missing some key details I do not believe it is ready for acceptance in its current form. If the manuscript was revised to improve clarity and add the key missing details (particularly the performance metric), I would be happy to change my recommendation to a weak accept. If the authors further incorporate an additional non diffusion baseline, I would change to a strong accept.

---

### Official Review · Reviewer_djYt · 2023-07-17

**Confidence:** 3
**Originality:** Fair
**Technical Quality:** Fair
**Clarity Of Presentation:** Poor
**Impact:** 2

**Recommendation:**

Weak Accept: I recommend accepting the paper, but will not argue for my recommendation if the majority of other reviewers have a different opinion.

**Review:**

Although the method is interesting, it appears to require a lot of work, engineering and prior knowledge to get it working for a novel skill. For example, consider the skill of pouring.

Firstly, all demonstrations of the pouring skill must be coded in simulation. This requires engineering a heuristic for the pouring skill that will be compatible with many different function and anchor object categories and will work for multiple pouring tasks, such as: pouring from a mug into a bowl, from a pan into a bowl, from a wine glass into a container etc. Having coded such a heuristic that works reliably enough to generate multiple demonstrations involving different object categories is already equivalent to coding a policy capable of reliably executing the skill, somewhat reducing the need for the proposed method.

Secondly, all demonstrations must be annotated with part-based functional correspondences. For example, for the pouring skill this would be <align, rim(mug), rim(bowl)>, <tilt, body(mug), body(bowl)> and <facing_up, handle(mug), body(bowl)>. Decomposing a general skill into these part-based functional correspondences is additional manual work that must be carried out for each new skill we would like to teach a robot.

Lastly, based on the part-based functional correspondences, the relevant object parts for the skill must be extracted in all demonstrated trajectories. Furthermore, during inference, they must also be segmented reliably.

**Strengths**:
- Simultaneously predicting a motion from different parts of a function and anchor object and combining the predictions is quite appealing.

**Weaknesses**
- The proposed approach requires a lot of engineering for each new skill. Namely, the biggest issue is providing demonstrations in simulation for a given skill involving different object categories for the anchor and function object. On the one hand, the data generation process could be automated. However, after doing this, we no longer have a need to learn the skill as we already have a policy capable of executing it. On the other hand, we could collect the demonstrations manually, which would be very time-consuming.
- The proposed approach requires a lot of prior knowledge, such as part-based functional correspondences for each skill and a separate frame of reference for each considered part of the object.
- Representing general skills as a starting and ending pose is quite limited. For example, for the pouring task, the robot may spill some of the liquid as it approaches the anchor object without any constraints on the orientation of the function object during the approach phase.

**Quality Of The Limitations Section:**

Limitations are not well addressed

**Questions For Rebuttal:**

**Training Data**:
- For each object category/instance, do you require a separate frame of reference for each of the considered parts of an object, e.g. for the rim, body and handle of a mug? If so, how do you obtain these frames of reference in the real world when encountering a novel object class?
- What is the exact form of the training data in the context of a single demonstration? Is this a (full) point cloud of each considered part of the anchor and function object, alongside with the final pose of the function object with respect to the anchor object? If the point cloud is full and not partial, how is this solved in the real world?

**Training and the model**:
- How many demonstrations are required to train the model?
- What is the architecture of the model?

**Experiments**:
- How exactly did you evaluate the methods in simulation? That is, how many evaluations did you do, how many object categories did you consider etc?
- Do you have any qualitative results for the real world experiment?

**Robotics Focus:**

Relevant but unlikely to deploy to hardware in near future

**Summary Of Paper:**

This paper proposes an approach for learning robot manipulation skills and generalising them to unseen objects from simulated demonstration data. The approach learns several diffusion models for a specific task, where each diffusion model is conditioned on only a single part of the function and anchor object and is responsible for capturing the distribution of valid function object poses for that task. For example, one diffusion model could be conditioned on the point cloud of the rim of a mug and bowl, while another could be on the bowl's body and the mug's handle. At test time, the output of all diffusion models is combined to predict the starting and ending pose for the function object, which, when realised, would complete the task.

**Summary Of Recommendation:**

The proposed method requires much manual engineering to set up a simulated pipeline for obtaining demonstrations. Moreover, the paper is very difficult to follow, and the experiments section is unclear, making it difficult to assess how well the method works and how it was evaluated. Moreover, it is difficult to understand how the method was deployed in the real world, given that each diffusion model makes predictions in the coordinate frame of the considered object part, which might not be annotated for novel real-world objects.

---

### Official Review · Reviewer_6mc4 · 2023-07-19

**Confidence:** 4
**Originality:** Very Good
**Technical Quality:** Very Good
**Clarity Of Presentation:** Very Good
**Impact:** 4

**Recommendation:**

Weak Accept: I recommend accepting the paper, but will not argue for my recommendation if the majority of other reviewers have a different opinion.

**Review:**


I think the general approach here is interesting. There are a couple of key ideas: 1) creating parts-based primitives rather than object-based primitives; 2) combining diffusion policies by simply adding. I think the parts-based approach is a little unusual in the literature and makes sense and I'm happy to see it here. Similarly, the approach to combining policies is also interesting. However, I'm not at all clear on how effective the adding approach would be -- the concern would be that multiple objectives cancel each other out. It would be good to explore this question in future work.

There is a comparison to PC-DDPM and part-aware PC-DDPM. I don't know if there are other methods that should be comared against, but it seems like at least some kind of baseline is being done here.

I'm also disappointed that the parts need to be hand labeled instead of being discovered. It does seem like there is a lot of hand coded information here. Nevertheless, the overall ideas are novel and I'm not super concerned about those points.

Overall, I think this is a decent paper that has interesting ideas.


**Quality Of The Limitations Section:**

Limitations are addressed clearly

**Questions For Rebuttal:**

None

**Robotics Focus:**

Sufficient demonstration on hardware

**Summary Of Paper:**


The paper proposes an approach to behavior cloning for manipulation. The manipulation behavior is generated by composing a small set of primitives. Each primitive is defined with respect to object parts, i.e. the rim of a bowl or the handle of a mug. The primitives are learned using behavior cloning. The system is presented with a database of expert trajectories that are labeled with behavior and with the parts involved. A diffusion model is trained to reproduce the behaviors, parameterized by the shape of the objects (encoded as a point cloud) and the task. A key aspect of the work is that multiple primtives can be combined at test time. This is done simply by adding the de-noising noise from all to-be-combined primitives. The resulting trajectories are expected to achieve the objectives of each primitive.


**Summary Of Recommendation:**

I think this is a strong paper, but probably not Earth shattering. I think we should accept, but if another reviewer is more knowledgeable and feels it should be rejected, I'm fine with that.

---

### Author Response · Authors · 2023-08-13
**General response to all reviewers**

We would like to thank all the reviewers for their insightful comments. We are glad everyone appreciates CPM’s novel approach. All the feedback has helped us greatly improve the paper.

Updates for the rebuttal:

- We added two baselines Transformer-BC [30] and TAX-Pose [37] from prior work. We are happy to report that when generalizing manipulation to novel object categories, our method outperforms Transformer-BC and TAX-Pose by 45.20% and 54.63%, respectively, for pouring, and 31.41% and 7.82%, respectively, for safe placing.
- We made significant changes to the paper to help clarify our method, evaluation, and limitation. We have attached the revised paper, with changes highlighted in blue.
- We added 6 new sections in the appendix to provide more detailed discussions about the network architecture, baseline implementations, statistics of the dataset, evaluation procedures, results with an alternative metric, and our assumptions. We hope these details help improve the credibility of the emprical results and reproducibility of our approach.

We highlight below some common themes across reviews:

**Combine multiple objectives.**
We want to highlight that each primitive diffusion model learns a generative distribution of the target transformations that satisfy a particular functional correspondence. Sampling from the combined diffusion model produces an “intersection” of these distributions, which correspond to actions that simultaneously satisfy all part-part correspondences.

**Other related methods.**
Besides emprically validated improvement in performance, we also want to emphasize that a key difference between CPM and prior methods is that CPM aims to learn manipulation skills that generalize to objects of novel categories. By contrast, existing methods for relational manipulation such as TAX-Pose and NDF focus exclusively on generalization to novel instances of the same category as training data. We aspire for our paper to inspire further research into manipulation with stronger generalization capabilities.

**Extracting parts.**
CPM relies on part labels for objects. These part labels are widely available in many 3D datasets, and for real objects, there exist part detection models such as [26]. This can be limiting in certain scenarios if the parts mentioned in the constraint description are less commonly annotated. In such cases, we believe extending the framework to automatically discover functional parts, either from scratch or by finetuning vision and language models such as [A1], are very exciting and important future directions.

How to effectively leverage the extracted parts is a key research question we aim to address. Compared to CPM, Part-Aware PC-DDPM does not enjoy the explicit decomposition of part functional correspondence terms. The experimental results confirm that merely providing the part information as additional inputs is less effective compared to our part-based correspondence formulation.

**Representing skills by two poses.**
We want to note that the proposed method is general with respect to the parameterization of the output trajectory. For example, the transformer-based diffusion model can be easily extended to output more waypoints to parameterize the trajectory if needed, even the entire trajectory. Here, we chose to represent trajectories by two poses primarily for sample efficiency concerns---training diffusion models for more waypoints in general would require more data. Furthermore, this parameterization is similar to "keyframe actions" used by recent learning-based maniuplation methods, for example, [A2, A3].

**Specification of symbolic constraints.**
We clarify that we only assume the “form” of such symbolic constraints associated with each task but not what these constraints mean. There is no need to specify any continuous parameters or mathematical formulas for these constraints. Because these constraints are symbolic, they can be easily specified by human users. We believe an interesting future direction is to leverage large language models to more efficiently extract constraints. Our method then learns the meaning of these constraints from data.

References:
- [30] Liu et al. StructFormer: Learning Spatial Structure for Language-Guided Semantic Rearrangement of Novel Objects. ICRA 2022.
- [37] Pan et al. TAX-Pose: Task-specific cross-pose estimation for robot manipulation. CoRL 2022.
- [26] Liu et al. Partslip: Low-shot part segmentation for 3d point clouds via pretrained image-language models. CVPR 2023.
- [A1] Zhang et al. Glipv2: Unifying localization and vision-language understanding. NeurIPS 2022.
- [A2] James et al. Coarse-to-fine q-attention: Efficient learning for visual robotic manipulation via discretisation. CVPR 2022.
- [A3] Shridhar et al. Perceiver-actor: A multi-task transformer for robotic manipulation. CoRL 2022.

---

### Author Response · Authors · 2023-08-14
**Looking forward to your feedback**

Dear reviewers, thank you once again for your valuable feedback which has helped us greatly improve the quality of our paper. We've addressed your questions in our earlier response and provided new results. As the end of the discussion period is approaching, we want to make sure we have addressed all of your concerns. Please let us know if you have any additional comments. Thank you!

---

### Decision · Program_Chairs · 2023-08-30

**Decision:**

Accept (Poster)

**Comment:**

This paper proposes a novel method based on object part decomposition and their correspondences to improve learning and generalization of robotic manipulation skills. All the reviewers recognize that the paper contains interesting ideas despite the fact that the current state of the approach requires a lot of manual engineering work. In the period of the rebuttal, the authors have provided careful answers with additional experimental results, e.g., two novel baselines,  to address the major questions of the reviewers. 3 reviewers suggest weak accept and one switches from strong reject to  strong accept after the rebuttal.